# A Survey on Model MoErging: Recycling and Routing Among Specialized Experts for Collaborative Learning

**Prateek Yadav**[*][1]   **Colin Raffel**[*][2,3]
**Mohammed Muqeeth**[5]   **Lucas Caccia**[6]   **Haokun Liu**[2,3]
**Tianlong Chen**[1]   **Mohit Bansal**[1]   **Leshem Choshen**[4,5]   **Alessandro Sordoni**[6,7]

[*] *Equal Contribution*

[1] *UNC-Chapel Hill,* [2] *University of Toronto,* [3] *Vector Institue*

[4] *MIT,* [5] *MIT-IBM Watson AI Lab,* [6] *Microsoft Research,* [7] *Mila*

*Correspondence Email: {praty@cs.unc.edu, craffel@gmail.com, alsordon@microsoft.com}*

**Reviewed on OpenReview:** *https://openreview.net/forum?id=u0azVc9Y0y*

## Abstract

The availability of performant pre-trained models has led to a proliferation of fine-tuned expert models that are specialized to a particular domain or task. Model MoErging methods aim to recycle expert models to create an aggregate system with improved performance or generalization. A key component of MoErging methods is the creation of a router that decides which expert model(s) to use for a particular input or application. The promise, effectiveness, and large design space of MoErging has spurred the development of many new methods over the past few years. This rapid pace of development has made it challenging to compare different MoErging methods, which are rarely compared to one another and are often validated in different experimental setups. To remedy such gaps, we present a comprehensive survey of MoErging methods that includes a novel taxonomy for cataloging key design choices and clarifying suitable applications for each method. Apart from surveying MoErging research, we inventory software tools and applications that make use of MoErging. We additionally discuss related fields of study such as model merging, multitask learning, and mixture-of-experts models. Taken as a whole, our survey provides a unified overview of existing MoErging methods and creates a solid foundation for future work in this burgeoning field.[1]

## 1 Introduction

The development of large-scale pre-trained models increasingly aims to create general-purpose AI systems that can perform any task without requiring task-specific training. Improvements in these models are often driven by scale, i.e. training a larger model on a larger dataset (Hestness et al., 2017; Kaplan et al., 2020). However, even with increased scale these models are not yet truly "general purpose" and often struggle with certain tasks and/or domains (McCoy et al., 2023; Ling et al., 2023; Kandpal et al., 2023a). Unfortunately, pre-training a new model in hopes of improving capabilities can be incredibly compute-intensive (Li et al., 2023; Workshop et al., 2022) and is therefore impossible for most of the research and practitioner community. In addition to high computational costs, it can be difficult to localize which parameters of the model might be more useful for a specific use case and adaptively improve performance or reduce computation based on that information (Pfeiffer et al., 2023).

Fortunately, it is often possible to make targeted improvements to a pre-trained model via fine-tuning (i.e. further training on a specialized dataset). In addition, parameter-efficient fine-tuning (PEFT) techniques (Ding

---

[1] We are maintaining a github repo with list of related papers and their taxonomy at https://github.com/pclucas14/awesome-moerging.

et al., 2022; He et al., 2021; Mangrulkar et al., 2022) further increase fine-tuning efficiency and decrease the cost of serving such specialized models. PEFT introduces small components like Low-Rank Adapters (Hu et al., 2022) or (IA)$^3$ vectors (Liu et al., 2022) that surgically modify the original model while adding a negligible amount of parameters. Due to their compact size, these specialized PEFT modules can be cheaply shared, facilitating the dissemination of an ever-growing number of adapters across various platforms. The effectiveness of fine-tuning, combined with the recent release of performant open-weight pre-trained models like Llama (Touvron et al., 2023) or Stable Diffusion (Podell et al., 2023), has fostered the creation and release of a multitude of fine-tuned *expert* models.

This proliferation of expert models has led to the development of methods for re-using such experts to improve performance or generalization. Central to these approaches are *routing* mechanisms that adaptively select relevant experts for a particular task or query. These methods have been referred to as "MoErging"[2] since they frequently share methodology and ideas with mixture-of-experts (MoE) models (Shazeer et al., 2017; Fedus et al., 2022; Du et al., 2022) and model merging (Matena & Raffel, 2022; Ilharco et al., 2022; Yadav et al., 2024). However, MoErging methods are distinct from MoE approaches that jointly train all the experts from scratch (Gupta et al., 2022). Instead, experts are provided by a distributed and decentralized community of contributors and are not trained by a centralized body. In addition, unlike merging methods that typically produce a static combination of models, MoErging methods adaptively combine models to improve performance on a per-query or per-task basis.

Model MoErging has several attractive properties compared to typical paradigms for monolothic model development. First, reusing and routing among independently-trained expert models can enable decentralized model development, alleviating the need for extensive centralized data and compute resources, while still ingesting a large amount of data and computation from different collaborators. Second, the modular use of expert models facilitates expansion of capabilities by adding experts or making localized updates by changing individual experts. Such changes are more "transparent" than standard model training by virtue of the fact that experts are often specialized specific functionalities. Finally, these systems can ideally generalize compositionally by identifying and "remixing" fine-grained skills from expert models in various ways, thereby extending their abilities beyond the experts' initially intended scope (Pfeiffer et al., 2023).

The promise of decentralized development of modular AI systems by recycling experts has led to an explosion of recent work on MoErging. The rapid pace of development in this budding subfield has often led to papers being unaware of and/or omitting comparison to one another. In addition, differences in assumptions, experimental setups, and problem settings across papers can further conflate comparison. Hence, the goal of this survey is to review the many recent publications and projects that can be considered a form of MoErging. To facilitate comparison and clarify assumptions, we categorize existing works in a novel taxonomy encompassing possible design choices at three distinct levels: i) the *experts* that are independently trained and shared by contributors in a decentralized fashion; ii) the *routing* strategy that determines how to select and aggregate experts; and iii) the downstream *application* targeted by end users. In addition, we discuss related lines of inquiry, tools that support this approach to model development, and areas for future work.

## 2 A Taxonomy for MoErging Methods

Broadly, our survey focuses on "MoErging", a new paradigm for decentralized model development that aims to recycle expert models trained asynchronously by distributed contributors. We can organize the stages and components of MoErging methods into three categories: (1) experts, (2) routing, and (3) application. The experts are the specialized models that are trained and shared by individual contributors. Importantly, experts are trained *independently*, i.e. the expert contributors do not have access to one another's compute or data. Once expert models have been shared, MoErging methods perform routing, which aims to select and aggregate the contributor-provided expert models in order to improve performance or generalization. To process a given query or adapt to a target dataset, routing can operate in various ways, for example: (1) adaptively select a single expert model, (2) route different examples or processing steps to different experts, (3) learn a layer to extract relevant information from all experts, and/or (4) combine the expert models in an adaptive way. Some MoErging methods assume that the expert contributors share not only their expert

---

[2]See e.g. https://huggingface.co/spaces/open-llm-leaderboard/open_llm_leaderboard

models but also their training datasets so that they can be used to design or create the routing strategy. Finally, the aggregate system is applied to some particular use case, e.g. processing a query or solving a target task. Different MoErging methods are designed for different use-cases, including zero- or few-shot adaptation to in-distribution or out-of-distribution task (i.e. a task for which there is or is not a trained expert model in the system).

While MoErging methods share an overarching goal, the large design space and range of possible use cases have led to a wide diversity of design choices across different methods. At the same time, seemingly disparate methods sometimes only differ in a single assumption or target application. In addition, most MoErging methods were developed over the past year or so. This contemporaneousness has resulted in studies rarely discussing or citing one another, further conflating comparison. To address this state of affairs, we introduce a taxonomy of MoErging methods that precisely enumerates the various design choices and use cases of MoErging. Our goal in designing this taxonomy is not only to elucidate the previously undocumented connections and commonalities among methods, but also to provide a framework to situate future work on MoErging. Our taxonomy is shown in Figure 5, with descriptions of each design consideration below.

## 2.1 Expert model design choices

MoErging involves recycling specialized expert models. Contributors of the expert models do their training independently, i.e. without access to one another's data or compute, and subsequently share their models. Design choices for the expert models include:

### 2.1.1 Expert Training

While contributors must train and share a model for it to be used as part of a MoErging system, a given MoErging method may further stipulate that the expert models are trained in a specific way. For example, PHATGOOSE (Muqeeth et al., 2024a) requires that expert model training includes an additional stage where gates are trained that are later used for routing. If a MoErging method stipulates a specific expert training procedure, we label it as **Custom**; otherwise, we label it as **Standard**. We note that many MoErging methods require access to *statistics* of each expert training dataset (e.g. each expert training set's average activation at some particular layer). We consider this a modification because it would not otherwise be done as part of standard expert training.

**Considerations:** The choice between **Standard** and **Custom** expert training procedures involves trade-offs between expert training complexity and better integration within the MoErging system. **Custom** training procedures, while potentially more complex to implement, can allow for better integration of experts with the routing mechanism, potentially leading to a better model (Muqeeth et al., 2024a; Wu et al., 2023; Diao et al., 2023). In contrast, **Standard** training leverages existing fine-tuning methods, and allows us to reuse trained experts available online (Huang et al., 2024b; Wang et al., 2024). However, it may necessitate more sophisticated routing and aggregation strategies to effectively utilize these off-the-shelf expert models.

### 2.1.2 Expert Data

A major motivation of the field of MoErging is to recycle the huge number of fine-tuned models being shared on model hubs. Such models are typically shared without their associated training data. However, certain MoErging methods assume access to expert training data, e.g. for learning the routing procedure. When expert data is shared, it is no longer a requirement that the experts must be trained independently. Furthermore, it would be possible to e.g. perform multitask training on all expert datasets simultaneously or carry out a modified expert training procedure. In the scenario where expert data needs to be shared, the benefits of MoErging methods are: (1) recycling of the compute required to train the expert models, and (2) decentralized training. In addition, apart from the reality that training data is often not shared alongside fine-tuned expert models, contributors may prefer to keep their training data private. We therefore categorize whether each method requires that expert training data is **Shared** or can remain **Private**.

**Considerations:** The choice between **Shared** or Private Expert Data fundamentally shapes MoErging methods. **Shared** data allows leveraging data embeddings for similarity-based routing (Chronopoulou et al.,

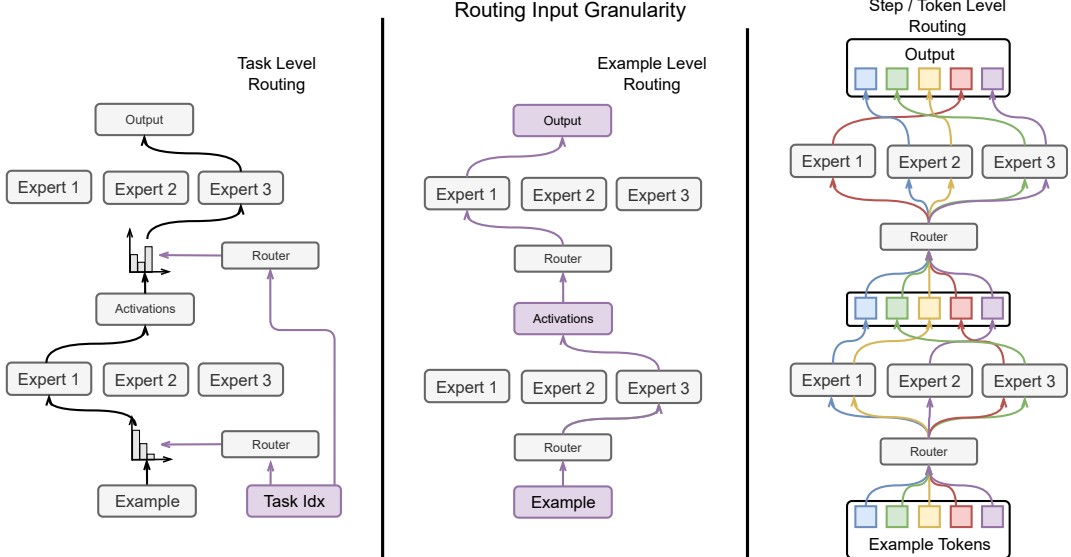

Figure 1: Different levels of granularity for routing decisions in MoErging methods. **Left:** Task Level Routing selects a single expert for all examples belonging to a specific task. **Middle:** Example Level Routing chooses an expert independently for each input example. **Right:** Step/Token Level Routing makes a routing decision (i.e., selects an expert) at each processing step or for each generated token. The purple elements indicate the input used by the router to make its decisions.

2023; Zhao et al., 2024b; Cheng et al., 2024; Jang et al., 2023), training a routing classifier (Lu et al., 2024a; Ong et al., 2025), and training module level routers on expert dataset distributions (Xu et al., 2024; Sukhbaatar et al., 2024), potentially leading to more informed expert selection (Zhao et al., 2024b; Bansal et al., 2024; Sukhbaatar et al., 2024; Zhao et al., 2024a). However, this necessitates data sharing, raising privacy and legal concerns with proprietary datasets. **Private** data methods are more broadly applicable and privacy-preserving, relying on open data sources (Shnitzer et al., 2024; Lu et al., 2024a; Ong et al., 2025) and expert models for routing and aggregation (Muqeeth et al., 2024a; Ostapenko et al., 2024). The core trade-off is between the routing utility gained from expert data and the practical limitations of data availability and privacy.

## 2.2 Routing design choices

In MoErging, expert models are collected to create an aggregate system to improve performance or generalization. A key step in this process is to create a "router" that can adaptively choose which model(s) should be used for a particular query or dataset. The creation of the aggregate MoErging system involves a large range of design choices, including:

### 2.2.1 Routing Dataset

To learn to route or select among expert models, MoErging methods often require a training dataset that we refer to as the "routing" dataset. Some MoErging methods make use of the **Expert**'s training datasets for the routing dataset, while others assume access to a **Target**-task dataset or a **General** dataset that covers a wide range of tasks. In addition, some MoErging methods do not explicitly train a router and therefore use **None**.

**Considerations:** The choice of Routing Dataset is critical as it dictates the data used to train the routing mechanism and significantly impacts the router's generalization capabilities and data requirements. Methods utilizing **Expert** datasets assume that the expert training data contains sufficient signal to learn effective routing for new inputs (Bansal et al., 2024; Zhao et al., 2024b; Sukhbaatar et al., 2024). This approach minimizes the need for additional data collection but may limit the router's ability to generalize beyond the

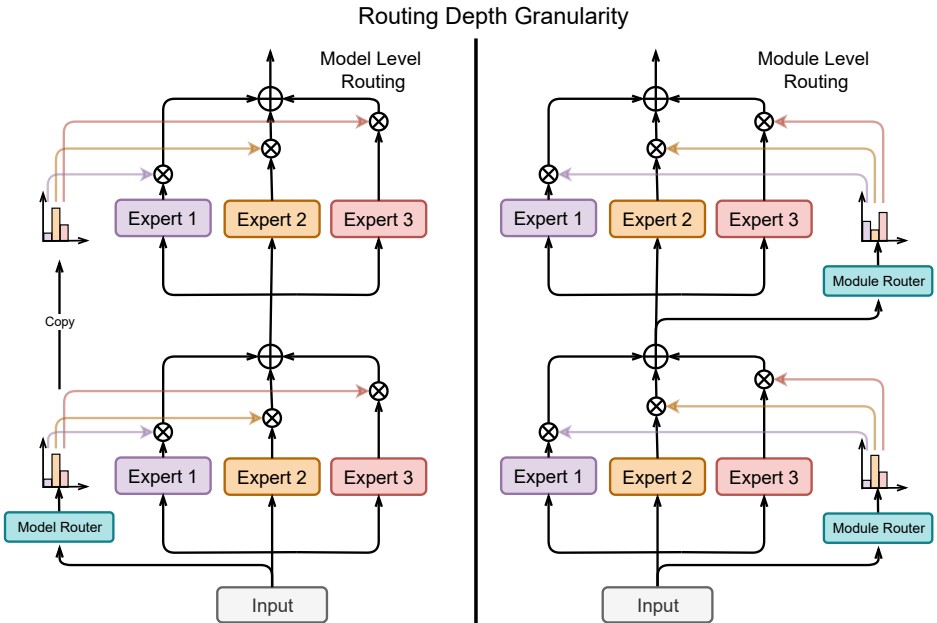

Figure 2: Different levels of granularity for routing depth in MoErging methods. **Left:** Model Level Routing applies a single routing decision to select experts that are then used across all applicable modules or layers of the model. **Right:** Module Level Routing makes independent routing decisions at each layer or module where experts are integrated, allowing for different experts to be active at different depths. The turquoise boxes represent the routers operating at either the model level or the individual module level.

distribution of expert tasks. Employing a **Target**-task dataset allows for optimizing the router specifically for the intended application, potentially leading to superior in-distribution performance (Tang et al., 2024c; Wang et al., 2024; Lin et al., 2024). However, this necessitates the availability of a labeled target dataset, which may not be feasible in zero-shot or few-shot settings, and may lead to overfitting to the target task, hindering broader generalization. Using a **General** dataset aims to train a more broadly applicable router, capable of generalizing across a wider range of tasks (Shnitzer et al., 2024; Lu et al., 2024a). These approaches rely on the assumption that the general dataset adequately represents the diversity of potential target tasks and requires access to such a diverse and representative dataset. Methods that use **None** for the routing dataset rely on heuristic routing mechanisms (Maxine, 2023; Durbin, 2024) or information stored from modified expert training (Jang et al., 2023; Chronopoulou et al., 2023; Muqeeth et al., 2024a), circumventing the need for router training data altogether. While data-efficient, the effectiveness of these methods hinges on the quality of the heuristics and their applicability to diverse tasks and expert models. The selection of the routing dataset thus involves a trade-off between data availability, task-specificity, and the desired generalization scope of the MoErging system.

### 2.2.2 Routing Input Granularity

Different MoErging methods make routing decisions at different levels of granularity. At the finest level, routing can be done per-**Step** (e.g. choosing a different expert model for each of a language model's generated tokens). In addition, routing can be performed once for each **Example** or query, or a single expert model can be chosen for all examples from some particular **Task**.

**Considerations:** Routing Input Granularity balances adaptability and cost, influencing methods' effectiveness. Higher frequency per-**Step** routing maximizes nuanced input adaptation but becomes computationally expensive and potentially unstable, with the risk of early routing errors propagating to later routing decisions (Muqeeth et al., 2024a; Shen et al., 2024). **Example**-based routing offers a balance (Zhao et al.,

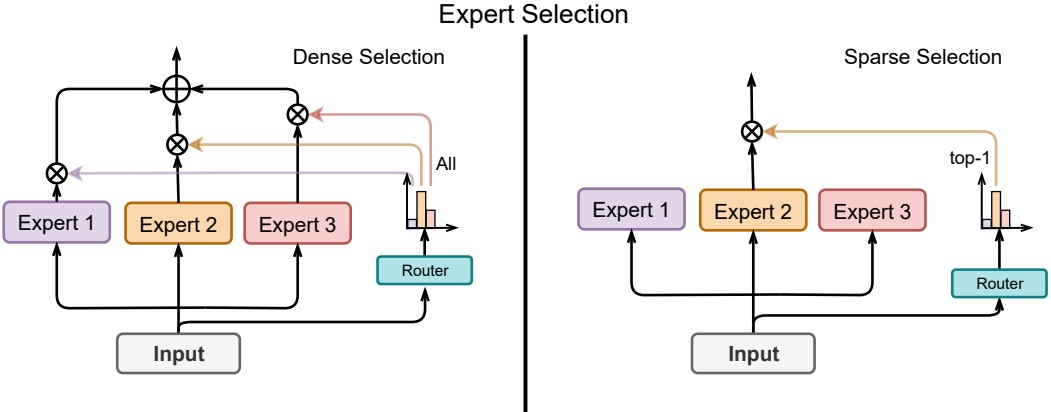

Figure 3: Different strategies for expert selection in MoErging methods. **Left:** Dense Selection utilizes the output of all available experts, often through a weighted combination. **Right:** Sparse Selection activates only a subset of the experts (e.g., the top-k most relevant ones) based on the router's decision. The router's output distribution indicates the selection strategy, with "All" implying dense selection and "top-1" implying sparse selection of the single most relevant expert.

2024b; Tang et al., 2024c). Lower frequency, **Task**-based routing prioritizes efficiency but assumes consistent expert needs across a task, potentially limiting adaptability for diverse task inputs (Huang et al., 2024b; Wu et al., 2023; Chronopoulou et al., 2023). Selecting the routing frequency involves trading off adaptability, computational efficiency, and the consistency of expert needs within and across inputs.

### 2.2.3 Routing Depth Granularity

Parameter-efficient fine-tuning methods like LoRA (Hu et al., 2022) or (IA)$^3$ (Liu et al., 2022) insert trainable modules at different layers throughout a model. Some MoErging methods therefore make per-**Module** routing decisions (i.e. with different routing decision at each layer where modules have been inserted, as in mixture-of-experts models (Shazeer et al., 2017)), while others make a single routing decision for the entire **Model**.

**Considerations:** Per-**Module** routing, operating at the level of individual modules or layers, allows for fine-grained control over expert utilization within different parts of the model (Diao et al., 2023; Ostapenko et al., 2024; Tang et al., 2024b). For instance, different experts might be selected for attention layers versus feed-forward layers. However, per-module routing significantly increases the complexity of the routing mechanism by making decisions at multiple depths, potentially leading to unstable routing as early routing errors can affect later routing decisions. **Model**-level routing, in contrast, applies a single routing decision to the entire expert model (Lu et al., 2024b; Ong et al., 2025). This simplifies the routing process, as only one decision is made per input or example, and reduces routing overhead. However, it assumes that a single expert (or combination) is globally optimal for all the layers of the model, potentially limiting adaptability. The choice between per-module and model-level routing thus involves a trade-off between routing granularity, architectural complexity, and the assumption of a few experts' relevance across model depths.

### 2.2.4 Expert Selection

When routing among experts, some MoErging methods make a **Sparse** selection (i.e. choosing only a subset of the experts) while others perform **Dense** routing (i.e. making use of all experts at once).

**Considerations:** Expert selection strategy governs the number of experts used, impacting computational efficiency and knowledge aggregation. Sparse selection (subset of experts) enhances efficiency by activating only the most relevant experts, crucial with large expert pools to avoid processing all experts (Muqeeth et al., 2024a; Xu et al., 2024). However, sparse selection requires precise routing to identify the relevant experts,

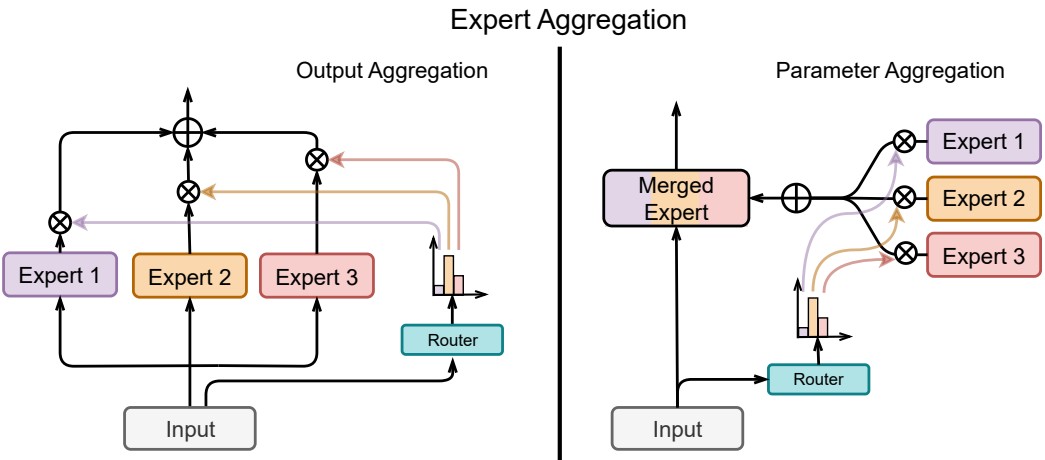

Figure 4: Different methods for expert aggregation in MoErging methods. **Left:** Output Aggregation combines the outputs of multiple selected experts, often using weights determined by the router. **Right:** Parameter Aggregation merges the parameters of multiple selected experts into a single, aggregated expert model before processing the input.

as any inaccuracy in selection can lead to information loss and reduced performance. Dense routing (all experts) maximizes knowledge aggregation potential and is advantageous when experts have overlapping, but complementary, skills for richer representations (Cheng et al., 2024; Zhao et al., 2024a). Dense routing is computationally expensive, particularly with many experts, and may degrade performance if irrelevant experts introduce noise and redundancy. Choosing between sparse and dense expert selection is thus a trade-off between computational efficiency, routing accuracy, and desired knowledge aggregation.

### 2.2.5 Expert Aggregation

If a MoErging method selects more than one expert, it must aggregate the experts or their outputs in some way. The **Expert Aggregation** method defines how the information from selected experts is combined, influencing the nature of knowledge integration and computational complexity. Aggregation methods include mixing the **Output** of experts, combining the expert's **Parameter** values before processing inputs, or **None** for methods that perform no aggregation (e.g. because they select a single expert).

**Considerations:** Expert Aggregation impacts knowledge mixing and computational complexity. **Output** aggregation flexibly combines expert predictions/representations by mixing their outputs (Lin et al., 2024; Bansal et al., 2024). Dynamic output combination is based on input relevance and can use techniques like weighted averaging, attention, or ensembling. Output aggregation performs post-expert processing allowing for independent expert operation. **Parameter** aggregation integrates knowledge at the parameter level by combining multiple expert parameters into a single aggregated expert (Lu et al., 2024b; Zhao et al., 2024a) – by using techniques like parameter averaging (Wortsman et al., 2022), merging (Yadav et al., 2024), or task vector arithmetic (Ilharco et al., 2022). Parameter aggregation yields a more compact model than output aggregation.

### 2.3 Application design choices

Once the expert models have been recycled into an aggregate system, users can then apply the system to their tasks or queries of interest. Different MoErging methods produce systems that support different usage patterns and incur different requirements on applications. Relevant design choices include:

### 2.3.1 Generalization

MoErging can aim to produce systems that improve performance on **In-Distribution** tasks (i.e. the tasks that the experts were trained on) or enable generalization to **Out-of-Distribution** tasks (i.e. those tasks for which there is no corresponding expert). However, many systems are applicable to both settings.

**Considerations:** The Generalization goal, whether **In-Distribution** (ID) or **Out-of-Distribution** (OOD), shapes design and evaluation criteria. For **ID** generalization, the primary objective is to enhance performance on tasks or domains that are similar to or within the distribution of the experts (Cheng et al., 2024; Sukhbaatar et al., 2024; Shen et al., 2024). Effective methods for ID generalization should excel at retrieving and aggregating relevant experts, potentially through fine-tuning routing mechanisms on ID data or leveraging expert specialization within the known distribution. However, ID-optimized methods may degrade **OOD** performance. In contrast, **OOD** generalization aims to enable performance on novel tasks (Wang et al., 2024; Muqeeth et al., 2024a; Mohammadshahi et al., 2024). Achieving OOD requires strategies that extrapolate expert knowledge, potentially via meta-learning, task similarity routing, or compositional generalization. OOD-focused methods prioritize robustness and adaptability over ID peak performance, and may require sophisticated routing and training. Many systems demonstrate applicability to both settings, highlighting potential for both specialized and general knowledge integration. The generalization goal guides design and evaluation, influencing routing data, training, and metrics.

### 2.3.2 User Dataset

MoErging methods may require a training dataset in order to be applied to a target task, which may be a **Few-Shot** dataset with a small number of labeled examples or a **Full** dataset with many labeled examples. Other methods require no target-task training dataset (i.e. they can be applied **Zero-Shot**). We make a slight misnomer and also refer to MoErging methods where an *unlabeled* target-task training dataset is required as "zero-shot".

**Considerations:** User Dataset availability dictates MoErging application and adaptation and impacts its usability and data needs. **Zero-shot** methods are appealing for their simplicity and speed, especially when labeled data is limited and costly. These methods inherently rely on the generalization capabilities of the pre-trained expert models and routing mechanism. Hence, without task-specific adjustments, their effectiveness can be limited in specialized downstream tasks. **Few-Shot** methods adapt the model using few labeled examples, often via prompt tuning or fine-tuning, balancing zero-shot usability and full fine-tuning performance with data-efficient task-specific optimization. The effectiveness of a few-shot method depends on the informativeness of examples and the adaptation technique. **Full** dataset methods use fully labeled data for end-to-end fine-tuning, potentially maximizing performance given sufficient data, but increase data and computational costs, reducing the data efficiency in some contexts. The choice of a dataset, thus trades off data needs, usability, performance, and applicability across data-scarce or data-rich scenarios.

## 3 Broad Categorization of MoErging Methods

Given the taxonomy in Section 2, we provide some high-level discussion about the commonalities and differences among MoErging methods. First, we note that many MoErging methods are remarkably similar, and that we can broadly group most of them into four categories based on how routing is learned: embedding-based methods, classifier-based methods, task-specific routing, router-free methods. Next, we explore these method categories, their strengths, weaknesses, data needs, and computational costs. These are crucial for practitioners seeking to apply MoErging effectively.

### 3.1 Embedding-Based Routing Methods

Embedding-based routing methods such as – AdapterSoup (Chronopoulou et al., 2023), Retrieval of Experts (Jang et al., 2023), Token-Level Adaptation (Belofsky, 2023), LoraRetriever (Zhao et al., 2024b), Mo'LoRA (Maxine, 2023), the embedding-based approach of Airoboros (Durbin, 2024), and Dynamic Adapter Merging (Cheng et al., 2024) – rely on comparing the input query embeddings with an embedding that represents

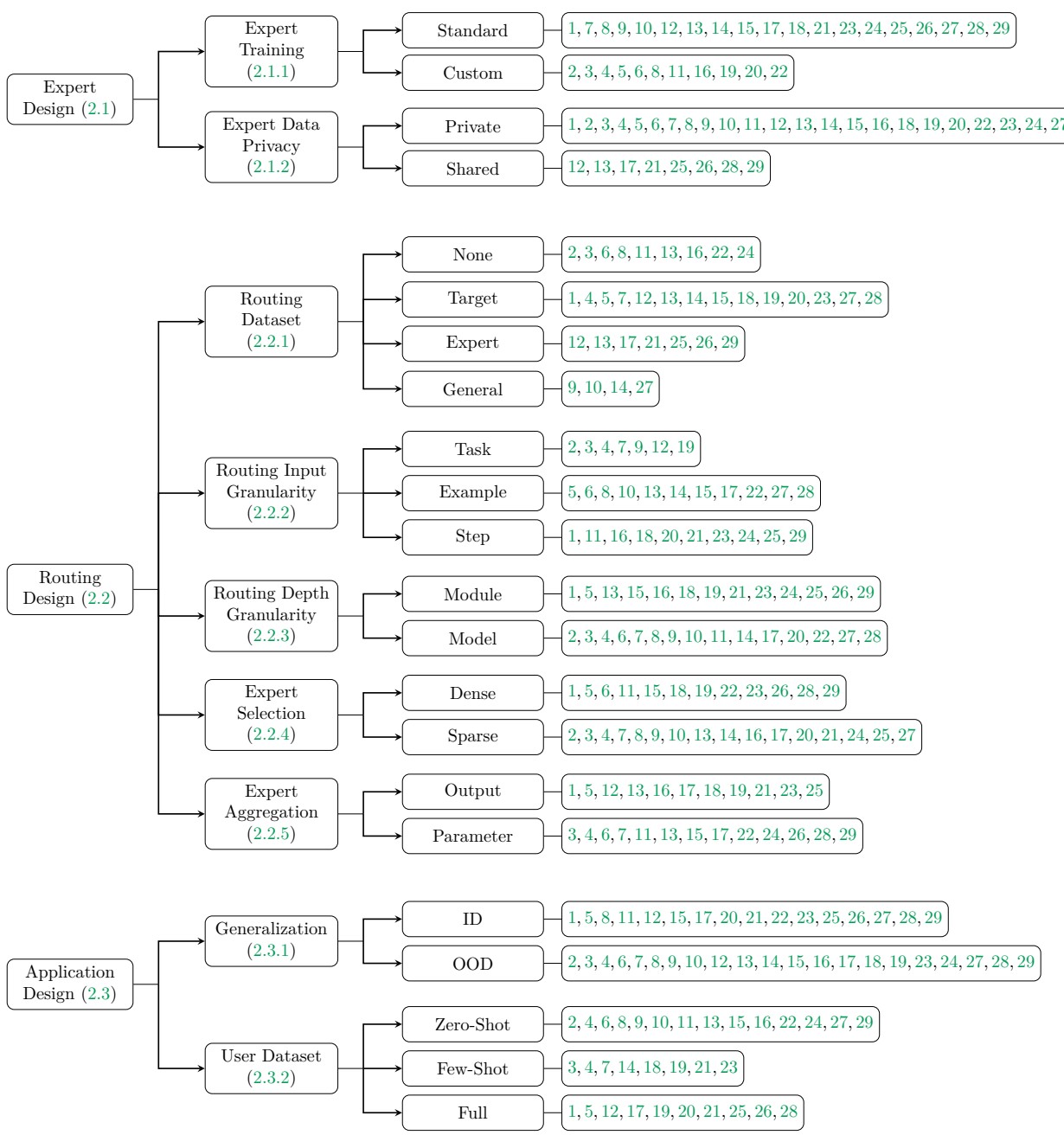

Figure 5: Taxonomy of model MoErging design choices. References in the leaf nodes link to sections for specific papers that make some particular design choice. We omit references to methods for which a given choice is not applicable.

the expertise of each available expert model. Often, these expert embeddings are derived from the experts training data, representative samples, or summary statistics thereof. Routing is then typically performed by selecting the expert(s) whose embeddings exhibit the highest similarity (e.g., cosine similarity) to the input embedding.

A key strength of embedding-based routing methods is their intuitiveness and minimal need for additional routing-specific training data. This makes them effective in zero-shot and few-shot settings and enables generalization to new tasks, provided the embedding space is capable of capturing the relevant task similarities. Moreover, efficient inference can be achieved through pre-computed expert embeddings and optimized similarity search techniques like FAISS (Douze et al., 2024). Furthermore, it is not necessary for all experts to follow the same architecture. However, their performance is critically dependent on the quality of the embedding space. A primary weakness is the reliance on pre-trained embedding models to capture nuanced task relationships. Suboptimal embeddings can result in poor routing, especially for semantically subtle, specialized, or out-of-distribution tasks where simple embedding similarity may not accurately reflect expert applicability.

Despite these limitations, embedding-based routers are well-suited for tasks where semantic similarity is a reliable performance indicator. They are most effective when expert training data distributions are relatively distinct, facilitating clearer separation in the embedding space. Conversely, they may falter in highly specialized domains where generic embeddings are insufficient, and where semantic similarity alone is inadequate for expert selection. They are also beneficial when expert models have different architectures.

### 3.2 Classifier-Based Routing Methods

Classifier-based routing offers an alternative to embedding-based methods by utilizing a dedicated, trained classifier to predict the optimal expert for each input. Methods like Zooter (Lu et al., 2024a), Branch-Train-Mix (Sukhbaatar et al., 2024), Routing with Benchmark Datasets (Shnitzer et al., 2024), Routoo (Mohammadshahi et al., 2024), and RouteLLM (Ong et al., 2025) exemplify classifier-based routing. Instead of relying solely on embeddings, these methods train a dedicated classifier to predict the optimal expert(s) for a given input for routing. This classifier is typically trained on labeled data, which could be the original expert training data (if shared), a general-purpose dataset covering a range of tasks, or a dataset specific to the target task. Downstream performance metrics can also serve as training signals, guiding the classifier to optimize expert selection based on desired outcomes. Effective training and performance hinge on the availability of sufficient, high-quality labeled data.

Classifier-based routing offers notable advantages, primarily in its capacity to learn and implement complex routing functions. Unlike embedding-based methods constrained by similarity measures, classifier-based approaches can capture nuanced and intricate relationships between input features and expert specializations. This enhanced complexity translates to potentially superior routing accuracy and a greater ability to balance performance and cost considerations within expert systems. The inherent flexibility of classifier-based routers further renders them more adaptable to a wider spectrum of tasks and domains, accommodating diverse expert architectures and application contexts. Classifier-based routing, while beneficial, faces inherent limitations. The foremost constraint is its data-dependent nature, which relies on labeled training data – a practical challenge in novel or unseen tasks where such data may be scarce or absent, thereby restricting generalization in zero- or few-shot learning scenarios. Furthermore, the dedicated classifier component increases computational requirements during both training and inference, enlarges the model size, and adds to the complexity of training. This added complexity is further affected by the routers architectural intricacy and the scale of the training dataset.

Classifier-based routing is best suited for data-rich scenarios with complex routing needs such as those explored in Routing with Benchmark Datasets (Shnitzer et al., 2024). They are ideal when nuanced expert selection beyond simple similarity is required, and can accommodate diverse expert designs across various applications given sufficient data. Practitioners are advised to adopt classifier-based routing when high precision is paramount and the target domain offers ample data, though its inflexibility in zero-shot or few-shot settings limits its broader applicability.

### 3.3 Task-Specific Routing Methods

Task-specific routing methods – like LoraHub (Huang et al., 2024a), LoRA-Flow (Wang et al., 2024), Adapter-Fusion (Pfeiffer et al., 2021), $\pi$-Tuning (Wu et al., 2023), Co-LLM (Shen et al., 2024), Weight-Ensembling MoE (Tang et al., 2024c), MoLE (Wu et al., 2024), MeteoRA (Xu et al., 2024), PEMT (Lin et al., 2024), MixDA (Diao et al., 2023), and Twin-Merging (Lu et al., 2024b) – are characterized by their focused approach to expert selection, learning routing strategies tailored for individual target tasks. Training is intrinsically linked to data from the target task or related datasets, often demonstrating data efficiency and requiring only limited task-specific examples to achieve good performance for the target task.

This focused training allows the router to become highly specialized for the nuances of a particular task, making it ideal for production environments where performance on a task is prioritized over adaptability. Often, only a relatively small amount of target-task data is needed to fine-tune the routing. Nevertheless, this specificity comes at the cost of generalization; without retraining, these methods cannot effectively handle new or diverse tasks, and their reliance on task-specific data may exclude them from applications where such data is unavailable. Computationally, training the router for each task can be time-consuming, particularly with large datasets or complex architectures, though the inference phase remains efficient once trained.

Task-specific routing methods are most appropriately applied when the primary goal is to maximize performance on a specific, clearly delineated task, and when some task-specific data is available. They offer a robust solution for targeted optimization in such contexts. However, practitioners should avoid these methods when broad generalization, adaptability to new tasks without retraining, or zero-shot capabilities are required. Their inherent trade-off necessitates a careful evaluation of application demands, prioritizing task-specific performance at the expense of wider applicability and flexibility.

### 3.4 Router-Free Approaches

Router-free methods offer an alternative to explicitly trained routers by directly determining expert choice through pre-existing mechanisms. Methods like Arrow (Ostapenko et al., 2024), PHATGOOSE (Muqeeth et al., 2024a), and LLM-based routing (Airoboros, LlamaIndex) which leverage expert properties, LLM capabilities, or precomputed information for routing. Arrow (Ostapenko et al., 2024) uses a zero-shot linear router with LoRA prototypes, while PHATGOOSE (Muqeeth et al., 2024a) employs sigmoid gates for token importance routing. Airoboros (Durbin, 2024) and LlamaIndex (Liu, 2024) depend on the inherent knowledge and reasoning abilities of pre-trained LLMs, and they only need descriptions of available expert models to perform routing. Critically, these methods bypass dedicated post-hoc router training by using precomputed gates or by using pre-trained LLM knowledge. This reliance on existing information significantly reduces data and computational demands compared to traditional router-based approaches.

The main advantage is that this design enables zero-shot functionality, allowing immediate deployment in settings where labeled data is absent, as demonstrated by Arrow (Ostapenko et al., 2024) and PHATGOOSEs (Muqeeth et al., 2024a) reasonable performance in initial testing phases. The absence of router training reduces computational overhead, with inference typically being fast despite potential precomputation costs for embeddings or LLM queries. These methods involve no training and do not require additional data. However, router-free methods may necessitate modifications to expert model training, increasing development complexity. Zero-shot routing can be inherently unstable, potentially leading to inconsistent performance. Furthermore, accurate expert retrieval for specific tasks, even when relevant experts exist, can be challenging. These factors highlight the need for careful implementation and thorough evaluation to mitigate potential drawbacks in router-free expert selection.

Router-free methods are particularly well-suited to applications featuring highly diverse queries and demanding zero-shot generalization to novel tasks. However, practitioners must carefully consider the inherent trade-off: while offering broad applicability through zero-shot capabilities, these methods may not achieve the task-specific performance attainable with dedicated routing schemes optimized for target tasks. Therefore, thorough evaluation remains crucial to fully characterize the nuanced performance of router-free approaches across diverse applications and to inform the future evolution of expert selection strategies.

### 3.5 Summary and Takeaways

Given these broad categorizations, we note that the differences between methods within a particular category frequently come down to the routing and expert granularity, the way experts are selected and aggregated, and the evaluation performed. We consider these differences to be relatively superficial compared to the way the router is built, which has significant implications in terms of what data is required and what settings a given method is applicable to. We therefore argue that methods within each category should, in most cases, compare to one another. However, such comparisons are rare in past work; for example, within the first category of embedding-based routers, only LoraRetriever compares to AdapterSoup. Comparison between groups would not make sense in some cases, though it could in principle be possible by swapping out the routing dataset of a given method (e.g., using a general dataset or expert datasets to learn the router in the third "task-specific router" category to enable comparison to methods in the other categories).

Apart from comparison to other methods, we reemphasize that assumptions about data access can make other often omitted baselines important. For example, if all expert datasets are assumed to be simultaneously available for the purpose of, e.g. training the router, then multitask training of the base model on the expert datasets should be considered as a baseline. More broadly, we consider data access to be a primary consideration in terms of which methods are applicable and/or realistic in different settings. For example, methods that require a labeled target-task dataset are, by definition, not applicable to improving zero-shot generalization, and methods that require that the expert training datasets are shared alongside adapters are not applicable to reusing the huge number of publicly shared adapters (which seldom have their training dataset released alongside parameter values). Next, we discuss many MoErging methods in detail and categorize them according to our taxonomy.

## 4   A Survey of MoErging Methods

Having established our taxonomy, we now provide a detailed survey of a few dozen recent papers that propose and study MoErging methods. Precisely delineating what is and is not a MoErging method is challenging because many past methods share the same basic motivation but differ in their application and framing. However, we believe that the papers we cover in this section provide a reasonably comprehensive overview of MoErging and MoErging-adjacent methods. Notably, most of the papers we discuss here only cite a small fraction of the other papers, suggesting that there is a general lack of awareness about relevant papers. Our survey aims to address this gap in knowledge.

For each method described in this section, we include an "infobox" cataloging the design choices made by each method according to our taxonomy. These infoboxes provide a point of reference to quickly understand each method and how it relates to others. However, there are cases where a given paper does not cleanly map onto our taxonomy. In such cases, we may denote that a paper considers **Multiple** options for a given design choice or that some design choice is **N/A** (not applicable).

### 4.1 AdapterFusion

| | | |
|---|---|---|
| **Expert Training:** Standard | **Expert Data:** Private | **Routing Dataset:** Target |
| **Input Granularity:** Step | **Depth Granularity:** Module | **Expert Selection:** Dense |
| **Expert Aggregation:** Output | **Generalization:** In-Distribution | **User Dataset:** Full |

Pfeiffer et al. (2021) propose a two-stage algorithm for sharing knowledge across task-specific adapters that consists of an extraction stage and a subsequent combination stage. In the extraction stage, the adapters (Houlsby et al., 2019) are trained independently on individual tasks. In the combination stage, a new fusion module is added to the top of all single-task adapters. The fusion module is a form of attention module (Vaswani et al., 2017), with its query from the input representation of adapters and the key and value from the output representation of the adapters. Then, the model trains only the fusion module parameters on a target task, therefore learning to combine all the individually trained adapters. Their experiment on 16 natural language understanding tasks shows in-distribution performance improvement on 12 tasks, compared to standard full model fine-tuning on the target task.

## 4.2 Retrieval of Experts

| | | |
|---|---|---|
| **Expert Training:** Custom | **Expert Data:** Private | **Routing Dataset:** None |
| **Input Granularity:** Task | **Depth Granularity:** Model | **Expert Selection:** Sparse |
| **Expert Aggregation:** None | **Generalization:** Out-of-Distribution | **User Dataset:** Zero-Shot |

Jang et al. (2023) argue that multitask training may underperform individually trained task experts equipped with a retrieval mechanism. Their proposed retrieval step encodes unlabelled examples from the target task, compares it to data encoded from each training task, and assigns each target datapoint to a specific trained expert. The expert with the most datapoints assigned to it is retrieved. Experiments are conducted using T0-3B and its associated training and evaluation sets (Sanh et al., 2022). This retrieval approach is shown to outperform T0-3B. Moreover, for certain benchmarks there exists a single oracle expert that performs significantly better than multitask training, showing the potential for better performance with a better retriever.

## 4.3 AdapterSoup

| | | |
|---|---|---|
| **Expert Training:** Custom | **Expert Data:** Private | **Routing Dataset:** None |
| **Input Granularity:** Task | **Depth Granularity:** Model | **Expert Selection:** Sparse |
| **Expert Aggregation:** Parameter | **Generalization:** Out-of-Distribution | **User Dataset:** Few-Shot |

Chronopoulou et al. (2023) combine different PEFT adapters trained independently over 21 website domains to enable few-shot transfer to novel domains. In order to select which domain adapters are the most relevant to the downstream task, the authors explore two approaches. The first uses a pretrained sentence-BERT (Reimers & Gurevych, 2019) representation averaged over 100 samples for each training domain and downstream task to compute a similarity metric. The second approach trains a gaussian mixture model using the representation of 100 samples from each training domain and then maps few-shot samples from the downstream task to their closest cluster. In either case, chosen adapters are retrieved and their parameter are averaged to produce an aggregate adapter for the downstream task. The authors show that both these approaches obtain better perplexity on 11 unseen domains than uniformly averaging all experts, and picking a single expert according to the same metrics.

## 4.4 $\pi$-Tuning

| | | |
|---|---|---|
| **Expert Training:** Custom | **Expert Data:** Private | **Routing Dataset:** Target |
| **Input Granularity:** Task | **Depth Granularity:** Model | **Expert Selection:** Sparse |
| **Expert Aggregation:** Parameter | **Generalization:** Out-of-Distribution | **User Dataset:** Multiple |

To transfer knowledge from similar tasks to a target task, Wu et al. (2023) make use of the Fisher Information Matrix (FIM)-based Task Vector method (Achille et al., 2019). Specifically, given a pool of adapters, they construct a new expert for a target task by finding the adapters whose FIM is among the top-$k$ most similar and averaging weights (including a target task-specific adapter) according to FIM similarity. The experts and their interpolation weights are jointly optimized to improve the target task loss. They also introduce a zero-shot variant, where the single adapter with the highest FIM is picked. Their results show improvement in multiple language and vision tasks.

## 4.5 MixDA

| | | |
|---|---|---|
| **Expert Training:** Custom | **Expert Data:** Private | **Routing Dataset:** Target |
| **Input Granularity:** Example | **Depth Granularity:** Module | **Expert Selection:** Dense |
| **Expert Aggregation:** Output | **Generalization:** In-Distribution | **User Dataset:** Full |

Diao et al. (2023) propose a two-stage algorithm to transfer knowledge from self-supervised domain adapters to target tasks. The first stage involves training domain-specific adapters with masked language modeling objectives on unlabeled data. In addition, a mean-square-error auxiliary loss is added to maintain the similarity between output representations of the domain adapter and the base model's feedforward network.

In the second stage, domain adapters are all added to the model and always activated. A series of MLP-sigmoid gates following the domain adapters control the weight to aggregate their outputs. This aggregated output is fed through a newly-introduced task adapter. Training in the second stage freezes the base model and domain adapters and updates the gates and task adapter.

## 4.6 Mo'LoRA

| | | |
|---|---|---|
| **Expert Training:** Custom | **Expert Data:** Private | **Routing Dataset:** None |
| **Input Granularity:** Example | **Depth Granularity:** Model | **Expert Selection:** Dense |
| **Expert Aggregation:** Parameter | **Generalization:** Out-of-Distribution | **User Dataset:** Zero-Shot |

Mo'LoRA (Maxine, 2023) considers the case where a base LLM (specifically, Llama 2) is being fine-tuned on a diverse general dataset (specifically, Wizard-EvolInstruct70k (Xu et al., 2023)). To train specialized models, the generalist dataset is first clustered based on embeddings produced by a sentence transformer (Reimers & Gurevych, 2019) and a LoRA is trained on each cluster. Then, the cosine distance between the embedding of a given query and the cluster centroids is used to produce a routing distribution. The parameters of the LoRAs are then averaged, weighted according to the routing distribution, and the query is processed using the aggregate LoRA.

## 4.7 LoraHub

| | | |
|---|---|---|
| **Expert Training:** Standard | **Expert Data:** Private | **Routing Dataset:** Target |
| **Input Granularity:** Task | **Depth Granularity:** Model | **Expert Selection:** Sparse |
| **Expert Aggregation:** Parameter | **Generalization:** Out-of-Distribution | **User Dataset:** Few-Shot |

Huang et al. (2024b) train one LoRA expert per task on a collection of 200 tasks from the Flan collection (Longpre et al., 2023), starting from the Flan-T5-Large as the base model (Chung et al., 2024). The experts are used to test few-shot generalization on a suite of 27 tasks from BIG-Bench Hard (Suzgun et al., 2022). LoraHub performs routing in two steps: first, 20 adapters are chosen at random from the full set of 200 training adapters; then, for each new task, the authors learn a fixed routing distribution over the randomly chosen adapters using a gradient-free method over a small task-specific training dataset. The routing probabilities are used to compute a weighted average of the chosen adapters' parameters to create a single specialized adapter. LoraHub is therefore focused on few-shot out-of-distribution tasks, i.e. it evaluates performance on a separate set of tasks but requires task-specific training data to learn routing weights.

## 4.8 Airoboros and LlamaIndex

| | | |
|---|---|---|
| **Expert Training:** Multiple | **Expert Data:** Private | **Routing Dataset:** None |
| **Input Granularity:** Example | **Depth Granularity:** Model | **Expert Selection:** Sparse |
| **Expert Aggregation:** None | **Generalization:** Multiple | **User Dataset:** Zero-Shot |

Airoboros (Durbin, 2024) is an open-source tool for generating and training on instruction tuning data. It includes functionality for selecting among a pool of expert models. Adaptive routing is supported in two ways: either by embedding 1,000 samples from each expert training dataset and retrieving the expert whose embedding is nearest to the query (composed of the system prompt and instruction) embedding via a FAISS index, or by asking an LLM which model to use for the query given a list of descriptions of each model. LlamaIndex (Liu, 2024) is an open-source library for connecting LLMs with data sources and other tools. Like airoboros, it includes functionality for building a model-level router by querying an LLM, with flexible choicses of the routing model and selection prompt.

## 4.9 Routing with Benchmark Datasets

| | | |
|---|---|---|
| **Expert Training:** Standard | **Expert Data:** Private | **Routing Dataset:** General |
| **Input Granularity:** Task | **Depth Granularity:** Model | **Expert Selection:** Sparse |
| **Expert Aggregation:** None | **Generalization:** Out-of-Distribution | **User Dataset:** Zero-Shot |

Shnitzer et al. (2024) reuse a collection of benchmark datasets (specifically HELM, Liang et al., 2022) to determine routing among LLMs on an unseen dataset. Specifically, they hold out one dataset while using the remaining datasets, called "benchmark data", for learning the routing. The evaluation is performed on all the LLMs in the pool on the benchmark data, and they define the correctness of each LLM for a given query with a binary score indicating whether the LLM can provide an acceptable answer to the given query. They embed the benchmark data using a sentence embedder, and for a query from the holdout dataset, the averaged correctness score from the $k$ nearest neighbors in the benchmark data is assigned as the score for this query and LLM. The average of all scores for all queries in the dataset is then taken to estimate how accurate an LLM is for the task. They propose three estimators: the first that takes the $\arg\max$ of the previously computed correctness scores over all the queries of the dataset. The second estimator applies a threshold on the correctness score of samples when averaging over queries in the dataset and accounts only for those that cross the threshold. To address out-of-distribution tasks, a third proposed estimator takes into account unlabeled out-of-distribution test samples by estimating the probability that the per-test-sample correctness score is accurate. The estimator defaults to the best LLM on the benchmark in cases of low confidence.

## 4.10  Zooter

| | | |
|---|---|---|
| **Expert Training:** Standard | **Expert Data:** Private | **Routing Dataset:** General |
| **Input Granularity:** Example | **Depth Granularity:** Model | **Expert Selection:** Sparse |
| **Expert Aggregation:** None | **Generalization:** Out-of-Distribution | **User Dataset:** Zero-Shot |

Lu et al. (2024a) propose Zooter, a learned router that aims to send each query to the best generalist language model (LM) within a pool of possible models. To train the router, predictions over a set of unlabelled instruction data are first collected for all LMs in the pool. The predictions are then scored by a reward model and the normalized scores across models are used as a training signal for the router. The router is kept relatively small (3 orders of magnitude smaller) compared to the LMs to keep routing cost low. Given the inherent noise in the scoring of queries using a reward model, the authors use a form of label smoothing: the reward for a given query is averaged with other queries with the same tags (e.g. "math", "creative writing") obtained from a pretrained tagger (Lu et al., 2023). When evaluated on generalist benchmarks like MT-Bench (Zheng et al., 2023) or FLASK (Ye et al., 2023b), Zooter performs similarly to naively routing each query to every LM the pool and selecting the response with the highest reward.

## 4.11  Token-Level Adaptation of LoRA Adapters

| | | |
|---|---|---|
| **Expert Training:** Custom | **Expert Data:** Private | **Routing Dataset:** None |
| **Input Granularity:** Step | **Depth Granularity:** Model | **Expert Selection:** Dense |
| **Expert Aggregation:** Parameter | **Generalization:** In-Distribution | **User Dataset:** Zero-Shot |

Belofsky (2023) formulate a routing approach for independently-trained LoRA experts. After training experts on a small set of specialized tasks, they form an expert representation by leveraging the experts' training data. Specifically, for each dataset, they compute the centroid of the embeddings of the dataset prompts. At test time, they normalize the cosine similarities between the embedding of the prompt generated so far and the experts' embeddings and combine expert parameters based on the resulting weights. The granularity of their routing approach is step-level and the routing decisions are shared across layers, with expert aggregation at the parameter-level. The evaluation is performed on *In-Distribution* tasks, i.e. they use the test set of the same tasks the experts have been trained on.

## 4.12  CALM

| | | |
|---|---|---|
| **Expert Training:** Standard | **Expert Data:** Multiple | **Routing Dataset:** Multiple |
| **Input Granularity:** Task | **Depth Granularity:** N/A | **Expert Selection:** None |
| **Expert Aggregation:** Output | **Generalization:** Multiple | **User Dataset:** Full |

Bansal et al. (2024) focus on composing knowledge from two models that can potentially have different architectures and sizes. Given an anchor model and an augmenting model, the goal is to have a final

model that is good at the anchor task, augmenting task, and a "composition" task that corresponds to the composition of the anchor and augmenting tasks. To achieve this, CALM adds multiple cross-attention layers between the augmenting and anchor model which takes in the input activation from both models. Then, the output from this learned cross-attention layer is passed on to the anchor model. Both the anchor and augmentation models are frozen and the cross-attention layers are learned in an end-to-end manner on a mixture of anchor and augmenting task in order to improve performance on the composition task. They use PaLM2-XXS as the augmenting model and use PaLM2-XS or PaLM2-S as the anchor model (Chowdhery et al., 2023). CALM is shown to be effective in experiments including adding low-resource support to an English model and improving the coding of the anchor model.

### 4.13 What the Weight?

| Expert Training: Standard | Expert Data: Multiple | Routing Dataset: Multiple |
| --- | --- | --- |
| Input Granularity: Example | Depth Granularity: Module | Expert Selection: Sparse |
| Expert Aggregation: Multiple | Generalization: Out-of-Distribution | User Dataset: Zero-Shot |

Holtermann et al. (2024) do not propose a new method for MoErging but instead introduce a framework under which they can perform experiments to better understand the various components and how they impact zero-shot compositional generalization. They frame such generalization as having three steps; (1) selecting a subset of experts, (2) deciding weights for each expert, (3) combining the different experts based on their weight. They experiment with five different types of scoring functions to select and weigh experts: uniform, sentence similarity, tf-idf, domain priors, and entropy. After selecting the scores and the experts they perform two different types of aggregation, parameter-level and ensembling the outputs. Their large-scale study produces various new insigts, including: ensembling generally yields better results than parameter averaging, good performance can be attained even with simple routing strategies, and that the number of chosen experts is more important than the precise weights assigned to them.

### 4.14 Routoo

| Expert Training: Standard | Expert Data: Private | Routing Dataset: Multiple |
| --- | --- | --- |
| Input Granularity: Example | Depth Granularity: Model | Expert Selection: Sparse |
| Expert Aggregation: None | Generalization: Out-of-Distribution | User Dataset: Few-Shot |

Mohammadshahi et al. (2024) describe a system that trains a router to perform model-level routing among generalist LLMs of varying sizes and architectures. A fixed budget is provided and the final objective is to maximize the overall performance across all queries while adhering to the budget constraints. Router training is done using a dataset of (query, response, evaluation score) triplets collected over many possible models. Mohammadshahi et al. (2024) use labeled target-task examples (specifically from MMLU) to synthetically generate the router training dataset with self-play for iterative refinement. However, we note that the method could in principle be used in zero-shot settings.

### 4.15 Weight-Ensembling MoE

| Expert Training: Standard | Expert Data: Private | Routing Dataset: Target |
| --- | --- | --- |
| Input Granularity: Example | Depth Granularity: Module | Expert Selection: Dense |
| Expert Aggregation: Parameter | Generalization: Multiple | User Dataset: Zero-Shot |

Tang et al. (2024c) argue that the interference when merging models should be dynamically resolved and hence design a MoErging method that averages all parameters except MLP layers which may contain more task-specific knowledge. They upcycle MLP layers into an MoE where each MLP from each expert model is converted to a task vector by subtracting the base model's parameters from the MLP layer's parameters. Routing is then performed between the expert MLP task vectors by multiplying the routing weights with the task vectors and then adding them back to the base MLP weight. Routing is done at the example level by taking the mean of all the token-level routing weights. Expert training data access is not required, but an unlabelled test dataset is used to learn the router by minimizing the routing distribution's entropy.

### 4.16 PHATGOOSE

| | | |
|---|---|---|
| **Expert Training:** Custom | **Expert Data:** Private | **Routing Dataset:** None |
| **Input Granularity:** Step | **Depth Granularity:** Module | **Expert Selection:** Sparse |
| **Expert Aggregation:** Output | **Generalization:** Out-of-Distribution | **User Dataset:** Zero-Shot |

Muqeeth et al. (2024a) focus on zero-shot generalization to unseen tasks by reusing existing adapters that are trained using a slightly modified training procedure. For each training task, they first train a LoRA module and then they add a sigmoid gate before each module which learns the importance of each token for this task. To compute this importance score they compute the sigmoid of the similarity between the gate and the per-token representations. Finally, they optimize the task loss for a given expert to learn these gates. Once LoRA and gates for all tasks are trained independently, then they create an MoE-style model from these experts for performing zero-shot generalization to unseen tasks. Specifically, for each Lora module, they create a router by stacking and normalizing all the gates from different experts. Then they normalize the token representation and route the token to the experts corresponding to the top-2 most similar gates. Results on improving the zero-shot generalization of T5.1.1 demonstrate that this approach outperforms other methods for learning post-hoc routing and can sometimes match the performance of explicit multitask routing.

### 4.17 LoraRetriever

| | | |
|---|---|---|
| **Expert Training:** Standard | **Expert Data:** Shared | **Routing Dataset:** Expert |
| **Input Granularity:** Example | **Depth Granularity:** Model | **Expert Selection:** Sparse |
| **Expert Aggregation:** Multiple | **Generalization:** Multiple | **User Dataset:** Full |

Zhao et al. (2024b) train a sentence embedding model to map from an input query into an embedding space that is then used to select an expert model to route the query to. The embedding space is constructed by sampling a subset of each of the expert model's training task and computing their average embedding. The embedding model is trained on a wide range of tasks in hopes of enabling generalization to unseen tasks and domains. Expert models are created as LoRA adapters to a base model, and are aggregated via merging or top-k output ensembling after routing is performed.

### 4.18 LoRA-Flow

| | | |
|---|---|---|
| **Expert Training:** Standard | **Expert Data:** Private | **Routing Dataset:** Target |
| **Input Granularity:** Step | **Depth Granularity:** Module | **Expert Selection:** Dense |
| **Expert Aggregation:** Output | **Generalization:** Out-of-Distribution | **User Dataset:** Few-Shot |

Wang et al. (2024) propose LoRA-Flow, which introduces a fusion gate at each layer of the transformer that processes the input at that layer and generates weights to perform a weighted average of the outputs from a set of pretrained LoRAs in the model. Some of these LoRAs are trained on multilingual language modeling, while others are task-specific and trained in English. These weights are generated for every token, making the routing token-level and layer-wise, with dense aggregation at the output level. Few-shot data from the downstream task of interest is used to learn this fusion gate, which comprises linear matrix and a bias vector. Experiments were conducted on math and coding abilities in a multilingual setting, specifically MGSM (Shi et al., 2022) for math and HumanEval translated into different languages for code. Their method outperforms LoraHub (section 4.7), which learns weights per task and averages LoRA parameters rather than outputs.

### 4.19 PEMT

| | | |
|---|---|---|
| **Expert Training:** Custom | **Expert Data:** Private | **Routing Dataset:** Target |
| **Input Granularity:** Task | **Depth Granularity:** Module | **Expert Selection:** Dense |
| **Expert Aggregation:** Output | **Generalization:** Out-of-Distribution | **User Dataset:** Multiple |

Lin et al. (2024) propose a method to train a parameter efficient adaptation for a new task by utilizing adapters from other tasks. First, for each source task, they train both a learnable soft prompt (Lester et al.,

2021) (initialized with a task description as in Raffel et al. (2020)) and an adapter. Then, for a target task, they initialize a soft prompt via an attention-style mechanism using the embedded target task description as a key and the source task prompts as keys and values. A task correlation matrix is constructed via a similar process, and a separate gating network is trained at each layer taking the correlation matrix as input. The gating network learns a weighting for computing an average of the source task adapters' outputs. Finally, a new target-task adapter is trained on downstream task data along with the gating network, soft prompt, and normalization parameters. This pipeline is shown to outperform other methods for recycling adapters such as SPoT (Vu et al., 2021) and ATTEMPT (Asai et al., 2022).

## 4.20 Co-LLM

| | | |
|---|---|---|
| **Expert Training:** Custom | **Expert Data:** Private | **Routing Dataset:** Target |
| **Input Granularity:** Step | **Depth Granularity:** Model | **Expert Selection:** Sparse |
| **Expert Aggregation:** None | **Generalization:** In-Distribution | **User Dataset:** Full |

Shen et al. (2024) trains a binary classifier on the top of a base model's last hidden state to determine when a base model should defer token generation to a frozen large model, facilitating collaboration between models. Given training data for a task, pseudo-labels are generated by evaluating both models and labeling instances where the large model predicts the correct next token while the base model does not. This data is used to train the classifier's parameters and is further used as initialization in the later stage when the classifier and base model are fine-tuned on the task. During inference, a threshold is set on the classifier using validation data that decides when to defer to the large expert model. This collaborative approach yields better results compared to fine-tuning the base model alone or using the frozen large model independently in instruction following, math, reasoning, and biomedical tasks.

## 4.21 Branch-Train-Mix

| | | |
|---|---|---|
| **Expert Training:** Standard | **Expert Data:** Shared | **Routing Dataset:** Expert |
| **Input Granularity:** Step | **Depth Granularity:** Module | **Expert Selection:** Sparse |
| **Expert Aggregation:** Output | **Generalization:** In-Distribution | **User Dataset:** Multiple |

Sukhbaatar et al. (2024) fine-tune each LLM on four different domains starting from a seed LLM (Llama 7B (Touvron et al., 2023)) to create expert LMs for each domain. They propose combining the FFNs of each expert LM to form an MoE, as in Lepikhin et al. (2020); Du et al. (2022), and averaging other parameters from each expert LM. The resultant model is fine-tuned on a training mixture corresponding to all the domains. During inference, top-2 routing is used at each MoE layer. They evaluate on downstream tasks in zero-shot and few-shot settings corresponding to each domain and find that their method performs comparably to the best domain expert LM for that task. Their method also performs comparably to a compute-matched counterpart, where the seed model is scaled to be similar size to the final model by upcycling (Komatsuzaki et al., 2022) and trained using multitask data.

## 4.22 Dynamic Adapter Merging

| | | |
|---|---|---|
| **Expert Training:** Custom | **Expert Data:** Private | **Routing Dataset:** None |
| **Input Granularity:** Example | **Depth Granularity:** Model | **Expert Selection:** Dense |
| **Expert Aggregation:** Parameter | **Generalization:** In-Distribution | **User Dataset:** Zero-Shot |

Dynamic Adapter Merging (DAM, Cheng et al., 2024) leverages domain-specific adapters of a base model to perform domain-incremental learning in the context of video question answering (VidQA). DAM first computes each domain-specific training set's average embedding from the penultimate layer of the base model. The distances between a given query input's embedding and the dataset average embeddings are then normalized to create a routing distribution. Finally, the query is processed by merging the domain-specific adapters using per-adapter weights set according to the routing distribution. On standard VidQA benchmarks, DAM significantly outperforms continual learning on the domain-specific datasets and nearly matches the performance of multitask training. However, the use of the base model to embed the input roughly doubles the computational cost.

### 4.23 Mixture of LoRA Experts (MoLE)

| | | |
|---|---|---|
| **Expert Training:** Standard | **Expert Data:** Private | **Routing Dataset:** Target |
| **Input Granularity:** Step | **Depth Granularity:** Module | **Expert Selection:** Dense |
| **Expert Aggregation:** Output | **Generalization:** Multiple | **User Dataset:** Few-Shot |

Wu et al. (2024) note that directly merging LoRA modules can degrade capabilities. They therefore aim to train routers to aggregate and reweight outputs from LoRAs at each layer where they have been introduced. Router training is performed on downstream data with the rest of the model (base model and LoRA parameters) fixed. In addition to a standard domain-specific loss, MoLE include a load balancing loss that aims to encourage the router to assign weight to all LoRAs. During inference, MoLE considers the cases where all LoRA outputs are used and where some LoRAs are manually removed. Experimental results include an analysis of performing model, layer, or module-level routing that demonstrates that module-level gating networks result in the best performance.

### 4.24 Arrow ↗

| | | |
|---|---|---|
| **Expert Training:** Standard | **Expert Data:** Private | **Routing Dataset:** None |
| **Input Granularity:** Step | **Depth Granularity:** Module | **Expert Selection:** Sparse |
| **Expert Aggregation:** Parameter | **Generalization:** Out-of-Distribution | **User Dataset:** Zero-Shot |

Ostapenko et al. (2024) explore methods to build and reuse a library of expert LoRAs for zero-shot task generalization. The proposed solution builds a MoE-like architecture, where the different experts are dynamically selected according to the input. To build a router in a zero-shot manner, the authors add a linear router at each layer, where expert prototypes are initialized to the top singular vector of a given LoRA expert. This then enables per-layer, per-step routing, using the top-4 experts at every selection step. The authors train experts on a 256-task subset of the FLAN dataset (Longpre et al., 2023), using Phi-2 (Microsoft Research, 2023) and Mistral-7B (Jiang et al., 2023) as backbones. They show performance gains both on in-distribution tasks, as well as on a collection of 10 held-out tasks, ranging from common sense reasoning to python programming.

### 4.25 MeteoRA

| | | |
|---|---|---|
| **Expert Training:** Standard | **Expert Data:** Shared | **Routing Dataset:** Expert |
| **Input Granularity:** Step | **Depth Granularity:** Module | **Expert Selection:** Sparse |
| **Expert Aggregation:** Output | **Generalization:** In-Distribution | **User Dataset:** Full |

Xu et al. (2024) propose an efficient method to dynamically select between multiple LoRA adapters. In each layer, a learned gating mechanism chooses a predetermined number of LoRAs to be activated for each token. The gating is learned by freezing the network and learning next token prediction over the same datasets used to train the experts. In addition to the architectural change, various engineering choices are made to ensure efficient parallelization of the gating choices, ultimately leading to substantial speedups. While MeteoRA is initially validated on in-distribution tasks with a labeled dataset, it could in principle be applied to out-of-distribution tasks.

### 4.26 PWE MoE

| | | |
|---|---|---|
| **Expert Training:** Standard | **Expert Data:** Shared | **Routing Dataset:** Expert |
| **Input Granularity:** N/A | **Depth Granularity:** Module | **Expert Selection:** Dense |
| **Expert Aggregation:** Parameter | **Generalization:** In-Distribution | **User Dataset:** Full |

Tang et al. (2024b) extend WE MoE (covered in section 4.15) to settings where Pareto-optimal performance is desired on a set of tasks. Task importance is set according to a user-specified "preference vector" (whose entries are nonnegative and sum to 1) that designates which tasks are more or less important. As in WE MoE, specialized models are upcycled (Komatsuzaki et al., 2022) into an MoE-style model by merging non-feed-forward network parameters via task vector arithmetic (Ilharco et al., 2022). Routers among the

feed-forward networks are trained by sampling random preference vectors and optimizing standard losses that capture Pareto optimality over the expert tasks. Routing based on preference vector-specified task weighting is shown to outperform merging methods that use the preference vector to set model weights.

### 4.27 RouteLLM

| | | |
|---|---|---|
| **Expert Training:** Standard | **Expert Data:** Private | **Routing Dataset:** Multiple |
| **Input Granularity:** Example | **Depth Granularity:** Model | **Expert Selection:** Sparse |
| **Expert Aggregation:** None | **Generalization:** Multiple | **User Dataset:** Zero-Shot |

Ong et al. (2025) aim to reduce inference costs while maintaining performance by dynamically routing each input to either a strong or a weak LLM. RouteLLM learns a router that estimates the probability that the stronger model will outperform the weaker one on a specific metric for a given input and select the weaker model if the probability is below a given threshold. The router is learned using a combination of preference data from Chatbot Arena (Chiang et al., 2024), instruction-tuning data such as Nectar (Zhu et al., 2023) and data with gold labels, such as MMLU (Hendrycks et al., 2020). RouteLLM is then evaluated on MMLU, MT-Bench (Zheng et al., 2023) and GSM8K Cobbe et al. (2021). Given that MMLU is both used for learning the routing and for evaluation, we described the Routing Dataset as Multiple, to denote that it comprises some examples from the target evaluation tasks and other generic tasks. The paper shows that RouteLLM can learn to choose between GPT-4 and Mixtral effectively, lowering inference costs. In some settings, it was observed that the inclusion of some target-task examples in the calibration data for the router was important to learn an effective router.

### 4.28 Twin-Merging

| | | |
|---|---|---|
| **Expert Training:** Standard | **Expert Data:** Shared | **Routing Dataset:** Target |
| **Input Granularity:** Example | **Depth Granularity:** Model | **Expert Selection:** Dense |
| **Expert Aggregation:** Parameter | **Generalization:** Multiple | **User Dataset:** Full |

Twin-Merging (Lu et al., 2024b) aims to address the gap in performance between a merged model and the original constituent models being merged. Specifically, Twin-Merging first constructs a "shared" model by performing task arithmetic (Ilharco et al., 2022) to merge all of the expert models. Then, "exclusive knowledge" experts are obtained by computing the rank-$r$ reconstruction from the SVD of the difference between the shared model and each expert model. Finally, a router is trained on a small task-specific training set to generate a weighting over the exclusive knowledge experts. The exclusive knowledge experts are then merged using the router's weights for a particular example. By performing adaptive merging, Twin-Merging is shown to create a single model that matches the individual models' performance.

### 4.29 RAMoLE

| | | |
|---|---|---|
| **Expert Training:** Standard | **Expert Data:** Shared | **Routing Dataset:** Expert |
| **Input Granularity:** Step | **Depth Granularity:** Module | **Expert Selection:** Dense |
| **Expert Aggregation:** Parameter | **Generalization:** Multiple | **User Dataset:** Zero-Shot |

Zhao et al. (2024a) extend LoraRetriever (see section 4.17) by including a router that outputs a distribution over the retrieved LoRA modules. The router is architected as a LoRA module that is trained on a subset of the expert tasks (in the same way as the LoraRetriever-based retrieval module). To ensure the router can handle out-of-distribution tasks, LoRA modules are randomly dropped out during router training. Routers are introduced at each location in the model where LoRA modules are introduced, producing per-token and per-module routing. Ultimately, performing a router-weighted average of module parameters is shown to outperform top-1 routing or unweighted module merging.

## 5  Related Methods and Problem Settings

Although recycling expert models for decentralized model development has only emerged in recent years, many established research areas are closely tied to this problem. Insights and solutions from these directions

may help accelerate progress on MoErging. In this section, we therefore outline several directions related to MoErging, discuss their commonalities and distinctions, and highlight promising connections.

## 5.1 Multitask Learning

Multitask learning (Caruana, 1997) aims to solve multiple tasks simultaneously, usually with a single model, to exploit commonalities across tasks. MoErging shares this high-level goal, but differs in that multitasking learning has mostly evolved in self-contained environments: one party decides on training data, trains the model, and deploys it for that party's own use. This condition leads to several key differences. First, multitask learning usually leverages relatively few tasks – either those the model developer has access to or those directly relevant to the developer's intended use. MoErging methods, on the other hand, aim to receive contributions from more tasks of greater variety and may target a wider range of use cases. Second, when a single party controls the entire data and model training pipelines, it is easier to implement sophisticated techniques. In contrast, for decentralized model development, innovations that deviate from the standard practice of sharing independently trained expert models may come at the cost of adoption.

Still, many previous works on multitasking learning tackle common challenges and can shed light on decentralized development. Multitask learning works often investigate the problem of task-relatedness (Standley et al., 2020; Bingel & Søgaard, 2017; Achille et al., 2019; Vu et al., 2020; Zamir et al., 2018; Mou et al., 2016), which is relevant to the routing mechanism. On the other hand, to better handle the commonalities and differences of multiple tasks, MTL has designed architectures that flexibly integrate task-specific components into the network (Misra et al., 2016; Ruder et al., 2017; Meyerson & Miikkulainen, 2017; Zaremoodi et al., 2018; Sun et al., 2019), which could be adopted to combine experts.

## 5.2 Mixture-of-Experts models

Mixture-of-experts (MoE) approaches train a set of experts that are sparsely activated during pre-training and inference (Jacobs et al., 1991; Jordan & Jacobs, 1994; Fedus et al., 2022; Shen et al., 2023; Shazeer et al., 2017). One of the main motivation for this approach is to scale model capacity while maintaining a fixed inference cost. In practice, experts are typically applied at the fully connected layers of the Transformer architecture (Lepikhin et al., 2020; Fedus et al., 2022). A sparse number of experts are typically activated per-token and per-layer by a learnable routing function. Switch Transformer (Fedus et al., 2022) only selects one top scoring expert and has demonstrated better scaling properties than previous work. Recent work applied mixture-of-experts also to attention layers (Shen et al., 2023). Given the discrete decision on which experts to apply, gradient estimation becomes difficult which makes training more challenging. Techniques such adding load balancing losses have been proposed to ensure all experts are used (Fedus et al., 2022). Recent work formulated improvements to sparse back-propagation that show promise in training better sparse MoE models (Liu et al., 2023). Overall, mixture-of-experts approaches generally focus on end-to-end "from scratch" training of experts and the router instead of learning post-hoc routing from independently trained experts as in MoErging. Kang et al. (2024) use the base LLM to generate synthetic data to create an MoE model by upcycling where each expert is specialized to some domain. MoErging may also introduce challenges that are distinct to those encountered when training MoE models from scratch – for example, it might be suboptimal for routing in a MoErged model to be balanced across experts, which could make it more challenging to distribute the model over many accelerators.

The pool of experts in an MoE-style model can be trained jointly in multitask settings. Restricting the size of the expert pool to be smaller than the number of tasks can enable similar tasks to share experts while avoiding interference across tasks with negative transfer. Polytropon (Ponti et al., 2019) and MHR (Caccia et al., 2023) learn a task-level router over experts, composing them into task-specific experts via a weighted average in parameter space. Task-level routing has also been explored in standard MoEs for more efficient inference (Kudugunta et al., 2021). Other routing strategies have also been explored in multitask MoE models. SMEAR (Muqeeth et al., 2024b) performs example-level routing by averaging expert parameters, while Zadouri et al. (2023) learns a per-token routing strategy. While MoErging methods generally assume expert models are trained independently, multitask MoE methods can be considered an important baseline when expert data is shared (in which case multitask training is possible).

MoE models have also been employed in continual learning to prevent forgetting by preserving important experts from previous tasks while updating or adding new experts for new tasks. For example, Chen et al. (2023) continuously trains an MoE model by freezing old feed-forward network (FFN) experts and incorporating new ones with KL regularization, outperforming dense continual pretraining. Aljundi et al. (2017) trains task-specific experts and uses autoencoders to identify the closest expert for new task and either fine-tunes it or adds a new expert initialized from the closest experts weights, demonstrating effectiveness in image classification and video prediction problems. Yu et al. (2024) uses Low-Rank Adaptations (LoRAs) as experts, employing a router per task to weight each expert and using autoencoders on pretrained input features to select the appropriate model for a new task. This approach also freezes the top-k experts from previous tasks to prevent forgetting. Li et al. (2024) offers theoretical analysis into the benefits of MoEs in continual learning on a sequence of linear regression tasks. Doan et al. (2023) suggests using model ensembles to reduce forgetting and adds loss terms to maintain linear mode connectivity between weights across tasks. While most MoErging methods do not explicitly target a continual learning setting, techniques for continual learning with MoEs could be applicable to MoErging methods (e.g. to continually update the router to account for a growing set of experts from the contributors).

## 5.3 Model Merging

Model merging aims to combine independently trained models that share an architecture into a single model that retains the individual models' capabilities. Merging can be performed via simple parameter averaging (McMahan et al., 2017; Stich, 2018) if the models fall into a linearly connected low-loss region in parameter space (Frankle et al., 2020; Wortsman et al., 2021). In the context of multiple expert models trained on different domains, (Ilharco et al., 2022) proposed task vectors, denoted as the parameter shift from the base model to the task-specific model and show that these can be combined to obtain a single generalist model. Several works go beyond uniformly averaging parameters, and explore various heuristics to estimate parameter importance. (Matena & Raffel, 2022; Tam et al., 2023) leverage the Fisher Importance Matrix to perform weighted merging. TIES-Merging (Yadav et al., 2024) resolve sign disagreemeent across experts and trim redundant values before merging. Ye et al. (2023a) uses a gating network trained on unlabeled data to predict task probabilities used to merge ViT layers and automatically select the appropriate classifier during inference. Merging can be considered a non-adaptive way of aggregating independently trained experts and is therefore a reasonable baseline for many MoErging methods.

Merging has also been applied to a continual learning setting where more expert models become available over time. Some efforts used such approaches to pretraining, slicing their data for efficient training (Li et al., 2022; Huh et al., 2024). Don-Yehiya et al. (2023) trained each expert on a different dataset to simulate the contribution of actual experts. While this line of work is similar to MoErging in terms of its focus on recycling a growing set of expert models, differences include that all models are being merged into one, and also that the new experts are already based on the previous model and are hence not completely independent.

## 5.4 Federated Learning

Federating learning (FL) is a widespread framework for collaborative learning where locally updated models trained on private user data are shared with a maintainer to improve a base model (McMahan et al., 2017). Some work has explored the intersection of MoE models and federated learning as a way to mitigate the data heterogeneity issue, where non-IID datasets can considerably diminish the accuracy of the FL model (Parsaeefard et al., 2021). FedMix (Reisser et al., 2021) and FedJETs (Dun et al., 2023) enable each client to construct an MoE with a shared router and a homogeneous local model. The router selects specific local models that are better adapted to a clients local data for output ensembling. However, they suffer from significant communication costs due to the model's distribution shifts.

In the personalized FL setting, `PEL-MoE` (Guo et al., 2020) customizes a MoE model by adapting the global model from the server and fine-tunes it to create a local expert model. These two models are then integrated with a local router into an MoE. During MoE training on the local data, the global expert remains frozen, while only the local expert and the router are updated to achieve personalization. More recently, `PFedMoE` (Yi et al., 2024) has introduced a sophisticated non-linear router network to extract more knowledge from local

data. This router network weights the final representations from each expert model, rather than their output labels like vanilla ensembling, in order to enhance the integration of global generalized and local personalized features. Work at the intersection of FL and MoE models is likely most relevant in cases where a user-tailored local model is desired and where user data must remain private.

## 6 Common Applications and Tools for MoErging

Development of an impactful machine learning model or a piece of software can happen in isolation. In contrast, MoErging methods provide a framework for decentralized development that relies on contributors for success. Consequently, appropriate infrastructure, systems, and tools are critical to the success of MoErging as a mode of decentralized model development. To facilitate this form of development, several open-source tools and platforms have emerged. These resources empower researchers and practitioners to experiment with different MoErging techniques, share their expert models, and collaborate on building more capable and robust AI systems.

Hugging Face has become a central hub for sharing pre-trained models and PEFT adapters. Initiatives like the Open LLM Leaderboard and AdapterHub (Beck et al., 2021) facilitate the discovery and comparison of models across various tasks and domains. Git-theta (Kandpal et al., 2023b) offers a decentralized alternative for tracking model weights, leveraging the Git version control software for collaborative development.

Several libraries have been developed to streamline the process of creating, combining and routing among expert models. Libraries like Hugging Face PEFT, Predibase's Lorax, Axolotl, and Unsloth can be used to efficiently create expert models either via full-finetuning or by parameter-efficient finetuning. These experts can then be used in libraries like Arcee's MergeKit (Goddard et al., 2024), Flow-Merge, Predibase's Lorax, Mergoo, and airoboros, each of which supports different forms of merging and MoErging. These libraries provide convenient interfaces for loading, managing, and aggregating expert models with different routing and aggregation strategies. Apart from these, ComfyUI offers a user-friendly graphical interface for designing and experimenting with complex merging pipelines, enabling users to visually connect and configure different expert modules to build performant aggregate systems.

The ongoing development of these tools and platforms highlights the increasing interest in MoErging and Merging for decentralized model development and its potential to democratize AI research and development. As the field matures, we can expect to see even more sophisticated tools and applications emerge, further empowering the community to collaboratively build and share more powerful and adaptable AI systems.

## 7 Takeaways and Open Problems

While there have been many papers developing MoErging methods that demonstrate improved performance or generalization, the field is still in its infancy. Notably, very few of the methods we surveyed are actually used in practice. We speculate that this could be due to a few factors: First, while many software tools that support MoErging exist (cataloged in section 6), many MoErging methods do not have user-friendly software implementations available. As such, applying MoErging often requires technical expertise that many users lack. Second, as we have highlighted in this survey, it is often unclear which MoErging methods are applicable and/or optimal for a given use case. More rigorous comparisons could be made through benchmarks, competitions, and empirical surveys like those developed for model merging (Tam et al., 2024b;a; Tang et al., 2024a). In addition, assumptions made by many MoErging methods (such as a custom expert training procedure and/or sharing expert training data) make them inapplicable unless the standard practice of solely sharing expert parameters changes. Moreover, as the field progresses, a crucial direction for future work lies in developing robust theoretical frameworks for MoErging. Such theoretical underpinnings are essential to formally understand the properties of different MoErging approaches, guide the design of more effective methods, and establish a deeper understanding of the conditions under which MoErging can offer substantial benefits. Finally, the promise of MoErging as a framework for collaborative learning may be held back by the lack of platforms that facilitate the various stages of a MoErging pipeline. For example, while the Hugging Face model hub provides an easy and widely used way to share models and datasets, it does not include extensive features for coordinating continual and communal model development.

If MoErging does ultimately become more widespread, there will be new challenges to tackle. For example, how can we identify (and possibly remove) redundant expert models? Or decide whether or not to add a new model to the pool? Or identify if a given model has been contributed by a malicious user who aims to degrade the aggregate system? In addition, adoption of MoErging will motivate the development of platforms that provide a "closed loop" for continual model development, where users can easily and cheaply fine-tune expert models and then share them to continually improve the aggregate system. In such cases, there will be multiple rounds of MoErging, raising additional research questions around whether MoErging can be used for continual improvement (or even pre-training) of a base model rather than one-off improvements on a single target task.

As the field of MoErging matures, it's likely that our survey and taxonomy will become out-of-date. In addition, while we aimed to be comprehensive, it is possible that we have missed certain methods or overlooked important design choices in our taxonomy. However, we hope that our survey provides a solid foundation and launch-off point for rigorous and coordinated research on MoErging.

### Acknowledgments

We thank Feng Cheng, Qian Liu, Jingwei Xu, Ziyu Zhao, Shannon Shen, Yong Luo, Omar Sanseviero, Jonas Pfeiffer, Shizhe Diao, Alireza Mohammadshahi, Rachit Bansal, Zhisheng Lin, Carolin Holtermann, Tal Shnitzer, Mikhail Yurochkin, Joshua Belofsky, Keming Lu, Xun Wu, Joel Jang, Alexandra Chronopoulou, and Ian Berlot-Attwell for feedback on our survey.

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
