# OpenReview forum: "A Survey on Model MoErging: Recycling and Routing Among Specialized Experts for Collaborative Learning"
_TMLR — Accepted by TMLR_

### Review · Reviewer_t29w · 2024-10-02

**Summary Of Contributions:**

This paper provides a comprehensive survey of Model MoErging, which shares concepts with MoE (Mixture of Experts) and model merging but introduces distinct ideas. MoErging methods adaptively combine models to improve performance on a per-query or per-task basis. This approach shows great potential, and the paper, as a survey, establishes a solid foundation for future research in this emerging field.

**Audience:**

Yes

**Broader Impact Concerns:**

No concerns.

**Claims And Evidence:**

Yes

**Requested Changes:**

While no major revisions are necessary, I offer the following suggestions:

- As mentioned, adding figures and math formulations would improve clarity.
- If available, providing theoretical foundations for MoErging methods is beneficial.

- It would be helpful for the authors to update the final version with recent publications, as this is an evolving field. Suggested papers include:

  [1] Hybrid LLM: Cost-Efficient and Quality-Aware Query Routing, ICLR 2024.

  [2] Harnessing the Power of Multiple Minds: Lessons Learned from LLM Routing.

  [3] RouterBench: A Benchmark for Multi-LLM Routing System.

  [4] RouterDC: Query-Based Router by Dual Contrastive Learning for Assembling Large Language Models, NeurIPS 2024.

**Strengths And Weaknesses:**

##### Summary of strengths

- The paper addresses a highly significant and timely topic in the field.
- The paper proposes a taxonomy that organizes and classifies various MoErging methods into three broad categories: expert design, routing design, and application design, along with several detailed subcategories. This framework offers a clear framework for comparison and understanding.
- The paper presents a thorough review of existing MoErging methods, covering numerous works in the field and related problem settings.

##### Summary of weaknesses

- The paper lacks visual illustrations and formalized equations.
- It does not include experimental results or analyses that compare the strengths and weaknesses of different methods.

---

> ### Author Response · Authors · 2025-03-13
>
> We thank the reviewer for their time and effort and apologize for the delay in response due to some unforeseen circumstances.
>
> **Weakness-1 and Requested Changes-1:** The paper lacks visual illustrations and formalized equations.
>
> **Response:** We have added Figures 1, 2, 3, and 4, visually illustrating key design choices like routing granularity, depth, expert selection, and aggregation, provides a more intuitive and accessible understanding of these concepts and improve the clarity, readability, and ease of understanding of the survey.
>
> We understand your about added math formulations for the papers, however, this was a conscious choice as: (1) It is very hard to come up with a unified notation for all the methods, (2) going into further mathematical details of each methods hinders the accessibility of the survey and takes away from the main point of providing a clear and concise overview to the readers and practitioners.
>
> While we maintain a degree of brevity to ensure broad coverage within a survey format, we have enriched Section 2, "A Taxonomy for MoErging Methods," in a way that significantly enhances its depth and practical relevance. Specifically, by adding the "Considerations" subsections (in blue text) after each design choice description, we move beyond mere descriptions of the choices themselves. These new sections delve into the trade-offs, limitations, and practical implications of each design decision, offering richer context and analytical depth.
>
> ---
>
> **Weakness-2:** It does not include experimental results or analyses that compare the strengths and weaknesses of different methods.
>
> **Response:** While this is a survey and not an experimental paper, we have added "Considerations" subsections to Section 2. These sections provide a literature-informed comparative analysis of design choice strengths and weaknesses, offering valuable insights without new experiments. Furthermore, Section 3 implicitly incorporates comparative analysis through its discussion of different broad categories of moerging methods.
>
> ---
>
> **Requested Changes-2:** If available, providing theoretical foundations for MoErging methods is beneficial.
>
> **Response:** We acknowledge the importance of theoretical foundations for moerging methods. We agree it is vital for the field, however, there is not much literature that explores these theoretical foundations. We have added a discussion on this in the "takeaways and open problems" section.
>
> ---
>
> **Requested Changes-3:** It would be helpful for the authors to update the final version with recent publications, as this is an evolving field ...
>
> **Response:** Thank you for pointing these paper, we commit to add these paper to the final version of the survey wherever appropriate. Additionally, to account for the every growing list of papers in this area, we plan to maintain a github "awesome moerging" repo with the categorization for all the new paper. We will add the link to that github repo in the abstract of the final version of the paper which we keep hidden now due to anonymity.
>
> ---
>
> Please let us know if you have any other questions and concerns or suggested edits and we will ensure to promptly incorporate those in the paper. Thanks!

---

### Review · Reviewer_TRBH · 2024-10-14

**Summary Of Contributions:**

The paper aims to review and categorize 29 MoErging Methods. There are also sections dedicated to broader context, tooling, and open problems.

**Audience:**

Yes

**Claims And Evidence:**

Yes

**Requested Changes:**

See above

**Strengths And Weaknesses:**

Strengths:
* The paper reviews and categorizes many methods, contextualizes them and provides open problems.
* The work has an encyclopedic structure, describing each method in a single paragraph with a couple of references and a categorization table. It allows for a quick recall or fetching of a relevant reference.

Weaknesses:
* The brevity of the description for each method limits the paper's ability to make it self-contained work.
* While reviewing a lot of methods, the paper does not necessarily guide with respect to the use of surveyed methods. This leaves the reader
who would like to use this survey for practically setting up their own pipeline with a sense of undertaste of wanting more. For instance, in Section 6, we can read
	* `the field is still in its infancy`,
	* `it is often unclear which MoErging methods are applicable and/or optimal for a given use case`,
	* `assumptions made by many MoErging methods (such as a custom expert training procedure and/or sharing expert training data) make them inapplicable`.

---

> ### Author Response · Authors · 2025-03-13
>
> We thank the reviewer for their time and effort and apologize for the delay in response due to some unforeseen circumstances.
>
> **Weakness-1:** The brevity of the description for each method limits the paper's ability to make it self-contained work.
>
> **Response:** We understand your point that the conciseness of the method descriptions might have limited the paper's self-contained nature. However, this was a conscious choice as going into further details into the methods hinders the clarity of the survey and takes away from the main point of providing a clear and concise overview to the readers and practitioners.
>
> While we maintain a degree of brevity to ensure broad coverage within a survey format, we have enriched Section 2, "A Taxonomy for MoErging Methods," in a way that significantly enhances its depth and practical relevance. Specifically, by adding the "Considerations" subsections (in blue text) after each design choice description, we move beyond mere descriptions of the choices themselves. These new sections delve into the trade-offs, limitations, and practical implications of each design decision, offering richer context and analytical depth. Furthermore, the inclusion of Figures 1, 2, 3, and 4, visually illustrating key design choices like routing granularity, depth, expert selection, and aggregation, provides a more intuitive and accessible understanding of these concepts, indirectly amplifying the impact of the method descriptions even if they remain concise in length.
>
> ---
>
> **Weakness-2:** While reviewing a lot of methods, the paper does not necessarily guide with respect to the use of surveyed methods. This leaves the reader who would like to use this survey for practically setting up their own pipeline with a sense of undertaste of wanting more. For instance, in Section 6, we can read
>
> **Response:** In our revision we have specifically focused on highlighting the similarities and differences, strength and weaknesses, and potential use cases of different methods.  The addition of the "Considerations" subsections in Section 2 and Section 3 are aimed at providing this practical guidance. These sections explicitly discuss when certain design choices might be more appropriate, the scenarios they suit best, and the inherent trade-offs and data requirements associated with each. This structured discussion of "Considerations," combined with the visual framework provided in the newly added Figure 1,2,3,4, now offers a concrete starting point for readers interested in setting up their own MoErging pipelines.
>
> While we acknowledge, as you quoted from Section 6, that the field is still in its infancy and definitive "optimal" methods are yet unclear, our survey, particularly with these added elements, now serves as a more robust guide for navigating the design space and making informed choices based on specific application needs and constraints.
>
> ---
>
> Please let us know if you have any other questions and concerns or suggested edits and we will ensure to promptly incorporate those in the paper. Thanks!

---

### Review · Reviewer_MJx4 · 2025-01-02

**Summary Of Contributions:**

This paper discusses "Model merging for collaborative learning", an important topic that allows decentralized model development and the ability to reuse expert models in new tasks. This paper comprehensively discusses and compares existing MoE merging work from three levels: expert design, routing design, and application design, so that the community can compare these methods fairly. However, this article still lacks in terms of the challenges in developing MoE Merging methods and the unified framework presentation of existing methods. In general, this paper still needs to undergo a revision to improve the overall quality.

**Audience:**

Yes

**Broader Impact Concerns:**

This article is a survey and does not involve questions about the impact of work ethics.

**Claims And Evidence:**

Yes

**Requested Changes:**

As discussed in the weaknesses section, the authors need to make further revisions. In addition, some additional revisions are as follows:
- This paper does not have any figs, which makes it difficult to read. The authors can consider adding some figures to clearly compare different MoE Merging methods and different routing design methods, so that readers can more easily understand the differences and connections between various methods.
- Some recent MoE-based model merging works [1-5] can be considered for citations and discussions. The authors need to further check other recent literature.
  - [1] Oh, Changdae, et al. "Adapting Foundation Models via Training-free Dynamic Weight Interpolation." Adaptive Foundation Models: Evolving AI for Personalized and Efficient Learning.
  - [2] Shen, Li, et al. "Efficient and effective weight-ensembling mixture of experts for multi-task model merging." arXiv preprint arXiv:2410.21804 (2024).
  - [3] Tang, Anke, et al. "Smile: Zero-shot sparse mixture of low-rank experts construction from pre-trained foundation models." arXiv preprint arXiv:2408.10174 (2024).
  - [4] Kang, Junmo, et al. "Self-MoE: Towards Compositional Large Language Models with Self-Specialized Experts." arXiv preprint arXiv:2406.12034 (2024).
  - [5] Ye, Peng, et al. "Merging Vision Transformers from Different Tasks and Domains." arXiv preprint arXiv:2312.16240 (2023).
- The authors need to update the published version information of the paper instead of citing the Arxiv version, for example, Retrieval of Experts [1] was published in ICML 2023. Adaptersoup was published in EACL 2023. Authors need to check the publication information of other important literature and update it.
  - [1] Exploring the Benefits of Training Expert Language Models over Instruction Tuning. ICML, 2023.
  - [2] Adaptersoup: Weight averaging to improve generalization of pretrained language models. EACL, 2023.

**Strengths And Weaknesses:**

This paper has the following advantages:
- This paper provides a detailed method classification framework, such as expert training methods and data requirements, routing’s data, depth, and granularity, and different downstream applications.
- This paper discusses the details of more than 20 related methods.
- This paper discusses the relationship between MoE merging and multiple fields (multi-task learning, model merging, etc.).

However, this paper still has the following weaknesses:
- This paper does not clearly explain the challenges/difficulties in developing a new MoE-based model merging method, and what aspects need additional attention. In addition, an in-depth discussion of which settings/processes of machine learning can be changed by model merging can further strengthen the necessity of model merging techniques.
- This paper overemphasizes the details of each method in Section 3, but lacks a macro comparison between methods. The author needs to add some systematic illustrations to more clearly explain the overall architecture/paradigm of a class of methods and how the design differences between methods are reflected.
- The author needs to further discuss in which settings MoE merging may be effective, in which settings it may fail, and why it is effective/ineffective.

---

> ### Author Response · Authors · 2025-03-13
>
> We thank the reviewer for their time and effort and apologize for the delay in response due to some unforeseen circumstances.
>
> **Weakness-1:** This paper does not clearly explain the challenges/difficulties ....
>
> **Response:** We have now explicitly addressed this in Section 6, "Takeaways and Open Problems," where we delve into several key challenges such as identifying redundant experts, addressing malicious contributions, and establishing platforms for continual MoErging development. Furthermore, within the newly added "Considerations" subsections in Section 2 (Taxonomy), we implicitly address challenges by discussing the trade-offs and limitations associated with each design choice, such as data requirements, computational complexity, and generalization scope. We believe this expanded discussion of challenges and limitations strengthens the paper by providing a more realistic and nuanced perspective on the field.
>
> ---
>
> **Weakness-2:** This paper overemphasizes the details of each method in Section 3 ...
>
> **Response:** We have added a new Section, "Broad Categorization of MoErging Methods," now explicitly groups methods into four categories based on routing learning approaches (embedding-based, classifier-based, etc.), facilitating a more structured comparison beyond individual method details. We discuss their similarities and differences, strength and weaknesses, and potential use cases where they might be a good idea.
>
> The Figure 5 containing the Taxonomy of MoErging Methods provide a systematic framework for understanding the high-level design choices and categories within MoErging, enabling a macro-level comparison across different methods. More, we have added Figure 1-4 to depict the various design choices at a high level. We believe these additions significantly improve the paper's ability to provide a macro-level perspective.
>
> ---
>
> **Weakness-3:** The author needs to further discuss in which settings MoE merging may be effective, in which settings it may fail, and why it is effective/ineffective.
>
> **Response:** We have added "Considerations" subsections throughout Section 2 (Taxonomy). Within these subsections, for each design choice, we now explicitly discuss "Suitable Scenarios," trade-offs, and limitations.
>
> Moreover, in the new "Broad Categorization of MoErging Methods" section, we also discuss the scenarios where different moerging methods might be effective/ineffective.
>
> ---
>
> **Requested Changes-1:** This paper does not have any figs ...
>
> **Response:** We have added Figures 1, 2, 3, and 4, visually illustrating key design choices like routing granularity, depth, expert selection, and aggregation, provides a more intuitive and accessible understanding of these concepts and improve the clarity, readability, and ease of understanding of the survey.
>
> ---
>
> **Requested Changes-2:** Some recent MoE-based model merging works [1-5] can be considered for citations and discussions ...
>
> **Response:** Thank you for pointing these paper, we would like to note that the first three papers [1,2,3] came out after this paper was submitted to TMLR.  For [4,5], they are not moerging exactly methods, hence we have cited them in the appropriate related work section of the paper.
>
> Additionally, to account for the every growing list of papers in this area, we plan to maintain a github repo with the categorization for all the new paper. We will add the link to that github "awesome moerging" repo in the abstract of the final version of the paper which we keep hidden now due to anonymity.
>
> ---
>
> **Requested Changes-3:** The authors need to update the published version information ...
>
>
> **Response:** Thanks, we checked this for all main papers in our survey and updated to their latest citations.
>
> ---
>
> Please let us know if you have any other questions and concerns or suggested edits and we will ensure to promptly incorporate those in the paper. Thanks!

---

### Decision · Action_Editor_fWe7 · 2025-03-18

**Recommendation:** Accept with minor revision

**Comment:**

Strengths:
- The survey is well-structured, providing a broad overview of MoErging techniques.
- It categorizes methods effectively and discusses their relevance to related areas such as mixture-of-experts (MoE) models and multitask learning.
- The paper introduces a useful taxonomy for organizing and comparing different model merging strategies.

Weaknesses:
- The paper lacks illustrative figures and equations that could improve conceptual clarity.
- It does not provide an explicit comparison of methods in terms of performance, trade-offs, or practical effectiveness.
- The discussion on open challenges and future directions could be expanded to offer more actionable insights.
- More details on applicable scenarios for different merging techniques would improve its usability for practitioners.

Some reviewers recommended major revision due to the lack of visual aids and comparative discussions. Others acknowledged the paper’s significance but suggested improvements to ensure a more structured and accessible presentation. The authors have made progress in addressing concerns by incorporating additional explanations and figures, but further refinements are needed to enhance clarity and completeness.

Given the importance of this survey and its potential to shape future research in model merging, I recommend minor revisions to strengthen the paper’s contribution. The authors should focus on adding visual illustrations, refining comparisons between methods, and enhancing the discussion on practical applicability. These changes will ensure the work meets the high standards of TMLR and maximizes its impact on the research community.

**Audience:**

The topic of model merging is timely and relevant, particularly as the field moves towards more efficient reuse of specialized models. The paper provides valuable insights into this growing area, making it beneficial to both researchers exploring expert model routing and practitioners interested in optimizing computational efficiency. However, the absence of comparative results and explicit guidance on the suitability of different methods for various tasks somewhat limits its practical impact.

**Claims And Evidence:**

The submission presents a comprehensive survey of model merging (MoErging) techniques, detailing various methodologies and their applications. While the paper effectively categorizes and contextualizes these methods, reviewers raised concerns regarding its clarity and completeness. Specifically, they noted the lack of visual aids such as figures and formalized equations, which could enhance understanding. Additionally, the survey does not sufficiently compare the effectiveness of different approaches or provide a unified framework that highlights key design trade-offs.